# Changes in Maximum Mandibular Mobility Due to Splint Therapy in Patients with Temporomandibular Disorders

**DOI:** 10.3390/healthcare10061070

**Published:** 2022-06-09

**Authors:** Bernhard Wiechens, Svea Paschereit, Tristan Hampe, Torsten Wassmann, Nikolaus Gersdorff, Ralf Bürgers

**Affiliations:** 1Department of Prosthodontics, University Medical Center Göttingen, 37075 Göttingen, Germany; schwea@gmx.de (S.P.); tristan.hampe@med.uni-goettingen.de (T.H.); torsten.wassmann@med.uni-goettingen.de (T.W.); nikolaus.gersdorff@med.uni-goettingen.de (N.G.); ralf.buergers@med.uni-goettingen.de (R.B.); 2Department of Orthodontics, University Medical Center Göttingen, 37075 Göttingen, Germany

**Keywords:** myofascial pain, TMD, splint treatment, electronic ultrasound device

## Abstract

Splint therapy is widely used in the treatment of myofascial pain, but valid studies on the efficacy of this therapy are rare. The purpose of the present study was to investigate which qualifiable and quantifiable effects of splint therapy are detectable. For this purpose, 29 patients (21 women, mean age 44.6 ± 16 years) diagnosed with myofascial pain (RDC/TMD) were investigated in this prospective clinical trial (10/6/14An). Patients were treated with Michigan splints applied overnight for three months. Before (T1) and after three months of treatment (T2), patients were registered with an electronic ultrasound device with qualitative and quantitative evaluation of the registrations and a qualitative assessment of pain symptoms using a verbal analog scale. Significant differences were found between maximum mouth opening (MMP) (*p* < 0.001) and right condylar movement (CM) at MMP (*p* = 0.045). Qualitative assessment revealed that 24 of 29 patients experienced an improvement in pain symptoms, 17 of whom experienced complete remission. The results of the qualitative and quantitative analysis provide indications of the effectiveness of the splint therapy. In addition to quantitative measurements, the ultrasound facebow technique was also able to provide qualitative information.

## 1. Introduction

Temporomandibular disorders (TMD) are a current and widespread problem in the general population, affecting about 5% to 31% of the population [1,2]. In addition, TMD is considered a major cause of pain in the head area [3,4], with an overall prevalence of about 46% over one’s lifetime [5]. According to the literature, the effects of myofascial pain in patients range from depressive states and low independence in activities of daily living to an overall lower health-related quality of life (HRQoL) [6,7]. TMD may be divided into many subgroups in which the causes of the dysfunction are joint associated or muscular associated [8]. The clinical expression of the complaints is therefore different in the case of muscular association (palpation pain with or without restriction of mouth opening) than in the case of joint association (dislocation with and without repositioning or with restricted mouth opening, arthralgia, arthrosis, arthritis) [9,10]. The Research Diagnostic Criteria for Temporomandibular Disorders (RDC/TMD) were established to precisely differentiate the symptoms, which are difficult to quantify, in order to enable uniform diagnosis against a background of high prevalence [10]. Despite generally valid and uniform diagnostic criteria, the therapeutic strategies published in the literature are diverse and range from analgesic or anti-inflammatory drugs, occlusal splints, physical therapies, trigger point injections, and acupuncture to Botox injections and lasers [11,12,13,14,15,16]. In this regard, a recent systematic review and meta-analysis explicitly reported the efficacy of all therapeutic approaches for TMD with exclusively a muscular origin [17]. Although 37.5% of the reviewed randomized controlled trials (RTCs) examined modern approaches such as laser therapy and found significant effect sizes, the authors cautioned not to disregard conservative rehabilitative approaches due to the paucity of studies. For occlusal splints, the authors summarized that high-quality RCTs are still needed to draw some conclusions, but at the same time noted the difficulty of designing a study with a sham or placebo treatment for splints; therefore, existing studies typically compare splints with another type of therapy or other splints. However, the most common therapeutic measures for painful TMD symptoms are occlusal splints, especially Michigan splints, which are designed to harmonize neuromuscular activity via centric positioning and simultaneous elimination of interfering contacts through achieving stable support of the dental arches by one another with balanced tooth contacts [18,19]. According to the original conception, Michigan splints are made for the upper jaw, from which deviations are only made in exceptional cases [19]. Based on the high clinical efficacy of occlusal splints, they are used frequently, although the actual quantifiable efficacy is not unanimous in the literature [11]. Furthermore, the essential aspect of pain reduction in this context is rarely reported with comprehensible qualitative data from the patients studied, which would possibly allow conclusions to be drawn about the relationships between therapeutic effects and simultaneous pain reduction [1]. The aim of the present study was, therefore, to evaluate the therapeutic effect of Michigan splints qualitatively in terms of pain reduction and quantitatively in terms of motion range improvement in patients diagnosed with myofascial pain (diagnosis I.a and I.b) according to the RDC/TMD criteria.

## 2. Materials and Methods

This longitudinal prospective clinical trial was approved by the Institutional Ethics Committee of the University Medical Center Göttingen (ethics number 10/6/14An) and executed in accordance with the principles of the Declaration of Helsinki. All patients participated in the trial on a voluntary basis after receiving comprehensive information about the aim and design of the study and signing an informed consent form. This report complies with the STROBE guidelines for observational studies.

Twenty-nine patients (21 women and 8 men, mean age 44.6 ± 16 years) with myogenic temporomandibular disorders (TMD) and clinical indication for occlusion splint therapy participated in the trial. Patients were treated by the Department of Prosthodontics at the University Medical Center Göttingen and were recruited between July 2014 and March 2016. Of the 32 patients initially screened, 3 female patients were excluded due to non-appearance for follow-up appointments (drop-out rate n = 3). Inclusion criteria were a minimum age of 18 years and the diagnosis of “myofascial pain” (diagnosis I.a and I.b) according to the RDC/TMD criteria (Research Diagnostic Criteria for Temporomandibular Disorders) [10]. Exclusion criteria were a history of former functional treatment (for example, occlusion splint therapy), edentulous patients, patients under orthodontic treatment, patients with syndromic underlying disease or craniofacial anomalies, nonspecific myalgias, arthralgias, and group II and III according to RCD/TMD Axis I.

To meet the basic inclusion criteria, a myogenic dysfunction had to be diagnosed according to the RDC/TMD Axis I diagnosis “I.a. Myofascial Pain” or “I.b. Myofascial Pain with limited opening” [10]. The diagnosis was made after detailed dental and functional examination. Specific medical history was obtained using the history questionnaire (RDC/TMD). An additional or sole RDC/TMD diagnosis of group II (disc displacement; DD) or group III (other joint diseases in terms of arthralgia, osteoarthritis, osteoarthrosis) led to study exclusion. All study patients had to have an additional clinical indication for sole occlusion splint therapy (Michigan splint), which was made by an experienced TMD practitioner (the last author).

After a TMD diagnosis for myofascial pain, impressions of upper and lower jaw dentition, centric bite registration, and facial arch registration were performed for subsequent Michigan splint fabrication in a dental laboratory. For centric condylar position registration, deprogramming of the masticatory muscles was performed, i.e., cotton rolls were placed between the dentition in an upright sitting and dorsally supported head position for 30 min. Immediately after removal of the cotton rolls, silicone was applied to the mandibular dentition for bite registration with Registrado X-tra (Voco GmbH, Cuxhaven, Germany). Subsequently, the individual position of the upper jaw in relation to the facial skull was registered by applying a facebow (AXIO-Quick III/ATB series transferbow, SAM Präzisionstechnik GmbH, Gauting, Germany). This was attached to a system-complementary articulator (Model 2P, SAM Präzisionstechnik GmbH, Gauting, Germany) by means of a mounting plate (SAM Präzisionstechnik GmbH, Gauting, Germany). After complete adjustment of the model casts in the articulator, an additional spacing of three millimeters was set starting from the zero position of the articulator pin. The bite spacing was thus accentuated in accordance with the concept for the subsequent fabrication of the Michigan splint according to the conventional procedure of Ramfjord and Ash [19], in a common workflow described by Patzelt [20]. Patients were asked to wear the splint every night during sleep, but not during the day.

The maximum mandibular mobilities in the frontal, horizontal, and sagittal planes of all 29 patients were recorded with an ultrasound electronic ARCUSdigma II device (KaVo Dental GmbH, Biberach, Germany) both before (T1) and three months after splint therapy (T2). A strict registration protocol was applied, where patients were examined in exactly the same examination environment at both T1 and T2. Patients were registered in the dental chair with strict adherence to a precise and reproducible sitting position, requiring the patients to be positioned with their shoulders and torso fully resting against the backrest and their heads supported dorsally against the headrest. Correct cranial positioning via the Frankfurt horizontal was checked throughout the entire measurement period, aligning the electronic facebow with the room plane. The application of the ARCUSdigma II measuring bow was always carried out according to the same scheme, with special regard to the manufacturer’s instructions. For calibration, the KaVo Transfer System (KTS, KaVo Dental GmbH, Biberach, Germany) was used to fix the individual reference points for software-integrated articulator programming. The mandibular transmitter unit was attached to a customized para-occlusal tray consisting of silicone applied to labial surfaces (Registrado X-tra, Voco GmbH, Cuxhaven, Germany) and a prefabricated system bite fork. In both measurements (T1 and T2), the patients were asked to consecutively perform first the maximum mouth opening, followed by the retrusion, as well as the left and right side laterotrusion under tooth contact. All movements began and ended in the maximal intercuspal position (MIP). The KaVo Integrated Desktop software (KaVo Dental GmbH, Biberach, Germany) delivered a function report (Function Report, ARCUS Digma II, KaVo Dental GmbH, Biberach, Germany) on the measurements (maximum opening, left condyle motion during maximum opening, right condyle motion during maximum opening, left condyle motion during retrusion, right condyle motion during retrusion, left laterotrusion, right laterotrusion) made after registration was completed. The content of this report was also outputted in tabular form via Microsoft Excel 2010 (Microsoft Corporation, Redmond, WA, USA) for analysis purposes.

For the quantitative evaluation of the collected measured values, statistical analysis was carried out, whereas the qualitative evaluation of the function reports was carried out visually. More harmonious and straightforward or more reproducible and symmetrical mandibular movements were evaluated as qualitative visual improvements. Less straight, limited, and asymmetrical movement patterns in the function report were considered to have no qualitative visual therapy result.

After 3 months of Michigan splint therapy, all patients were asked for subjective assessment of the therapy success in addition to the final electronic recording of mandibular mobility (T2). The patients were asked about subjective current pain sensation and how facial and jaw pain had changed after splint therapy. All patients were assessed by means of a verbal analog scale, with the following responses being distinguished: (1) no change from baseline, (2) improvement from baseline, (3) complete remission, and (4) exacerbation from baseline.

Data were transferred from the KaVo Integrated Desktop software (KaVo Dental GmbH, Biberach, Germany) to Microsoft Excel 2010 (Microsoft Corporation, Redmond, USA) for processing the raw data and then fed into SPSS Statistics Version 24 software (IBM Corp., Armonk, NY, USA) for statistical analysis. Metrically scaled variables were first tested for normal distribution using Kolmogorov–Smirnov and Shapiro–Wilk tests. Based on a non-given normal distribution, for descriptive statistics, metric variables were presented as median, minimum, and maximum as well as interquartile range (IQR). The significance level of all tests was set at *p* ≤ 0.05. Examination of the dependent variables for significant differences between the two dependent samples (same patient(s) at two measurement time points) was performed using the nonparametric Wilcoxon test. For final analysis of possible gender-specific differences (independent samples) regarding the variables examined, the Mann–Whitney U test was applied.

## 3. Results

### 3.1. Maximum Mouth Opening

To identify any therapeutic effect of occlusion splint therapy on maximum mandibular mobility, the digitally recorded maximum mouth opening values (based on incisal point movements in the sagittal plane) for all study patients (n = 29) were compared before (T1) and 3 months after continuous splint therapy (T2). Values with a positive sign indicate an increase in mobility and those with a negative sign indicate a reduction in mobility. An increase in the range of motion during mouth opening was observed for every single test patient. The median increase in the range of motion was 6.15 mm, with minimum and maximum values of 0.2 mm and 17.01 mm, respectively (Table 1).

A highly statistically significant difference (*p* < 0.001) was found for the collected dependent samples and distinct indications for a therapy-induced increase in the range of motion could be assumed.

### 3.2. Condyle Motion during Maximum Mouth Opening

In addition to the movement of the incisal point of the mandible according to Posselt’s envelope of motion [21], the respective movement patterns of both condyles were also recorded independently before and after therapy during maximum mouth opening. A median increase in range of motion of 1.93 mm at the left condyle and 2.73 mm at the right condyle was found, with minimum and maximum values of −1.29 and −0.75 mm and 10.55 and 18 mm, respectively (Table 1). From this, it could be concluded that most patients showed an increase in the range of condyle motion, but isolated reductions were also observed (Table 1). The applied Wilcoxon test revealed significant differences (*p* = 0.045) regarding the range of motion when considering the right but not the left condyle (Table 1).

### 3.3. Lateral and Retrusion Movements

Retrusion was analyzed using the left and right temporomandibular joints. The median increase in range of motion was 0.38 mm on the left condyle and 0.28 mm on the right condyle, with minimum and maximum values of −1.15 and −0.06 mm and 2.29 and 1.19 mm, respectively (Table 1). The majority of patients showed an increase in range of motion, which, however, did not reach the significance level. The examination of laterotrusion to the left and right also showed median increases in range of motion of 0.73 mm (left) and 0.63 mm (right). Minimum and maximum values were −1.05 and −3.90 mm and 4.73 and 5.11 mm, respectively (Table 1). Again, the majority of patients showed an increase in range of motion, which, however, did not reach the significance level.

### 3.4. Pre-and Post-Therapeutic Qualitative Assessment of Movement Patterns

After the objective assessment of the pure values from the mandible motion analyses, a descriptive comparison of the motion trajectories before and after therapy was performed. In the majority of patients, a clear harmonization of the movements (of the incisal point and the condyles) could be observed after splint therapy in the function reports (ARCUSdigma II). This harmonization was manifested in a lengthening and leveling of the trajectories of movement and an approximation to the classical movements of Posselt’s envelope of motion in the sagittal and transversal plane [21]. The recorded mandibular movements of an exemplary patient before and after splint therapy are given in Figure 1.

Finally, the entire study population was evaluated regarding recognizable harmonization of the movement patterns (Table 2), whereby both the qualitatively visually ascertainable degree of harmonization of the movement sequence and the quantitative metric changes were considered.

A comprehensive improvement in function was observed for 18 patients (Table 2), i.e., 62% of the total study population, as at least three variables were rated a “+”; thus, ≥50% of the variables showed both qualitative and quantitative improvement. Seven patients, i.e., just under 25% of the study population, showed no more than two “−”-rated variables, with only 33% of the variables showing no change and 66% showing either a solitary or full improvement in the variables. The smallest group consisted of four patients, i.e., 14% of the study population, for whom no qualitative or quantitative improvement was observed, with ≥3 “−”-rated variables.

Analyzing the function reports in detail, the five refractory patients showed a smaller increase in maximum mouth opening. With a median pre-therapeutic mouth opening of 49.6 mm ± 14.6 mm, a movement increase of 2.6 mm ± 3.4 mm, and a post-therapeutic mouth opening of 50.4 mm ± 12.8 mm, the patients achieved higher mouth openings compared with the 24 patients with improved or resolved pain symptoms, but a considerably lower rate of change. The remaining patients showed a maximum mouth opening of 42.5 mm ± 9.5 mm pre-therapeutically and a maximum mouth opening of 47 mm ± 8.6 mm post-therapeutically, resulting in an overall median movement increase of 6.7 mm. The refractory patients thus showed only 38% of the increase in mobility of the patients with improved or resolved pain symptoms.

### 3.5. Post-Therapeutic Qualitative Assessment of Pain Perception

The visual analog scale of the patients revealed unchanged pain symptoms in five patients and noticeable improvement in 24 patients, where 17 were completely pain free concerning the masticatory muscles and temporomandibular joints at re-evaluation. None of the respondents indicated worsening pain (Table 3).

## 4. Discussion

The present study showed a significant increase in maximum mouth opening and right condylar motion in patients with diagnosed myofascial pain (RDC/TMD I.a and I.b) after three months of Michigan splint therapy. Furthermore, 24 of the 29 patients reported a significant decrease in pain symptoms, and 17 patients experienced complete remission according to verbal analog scale assessment. None of the patients showed a worsening of symptoms. When classifying these results, it must be discussed that the use of scales involves individual and qualitative data from the patients. Not least for this reason, an adjuvant quantitative analysis of the treatment success was carried out. However, current studies on test reliability, validity, and minimal detectable change as well as correlations of demographic variables with comparable scales come to the conclusion of an excellent test reliability of comparable scales [22]. Furthermore, with regard to the five patients who experienced no improvement in the initial condition, it can be assumed that possibly no isolated myofascial pain in the sense of the RDC/TMD diagnosis I.a and I.b was present; thus, the initial therapy concept may not have been suitable for the cause of pain, emphasizing the difficulty of diagnosis. Taking into account the findings of Al-Moraissi et al. [11] and Deregibus et al. [1], who found a currently weak and inconclusive literature on the efficacy of splint therapy, the results of the present study provide a distinct indication of the therapeutic effects. The majority of existing studies approached the difficulty of capturing reliable outcomes by using surface electromyography (sEMG) derivations to formulate the therapeutic effects of splint therapy based on muscle activity changes with concomitant pain reduction [1,23,24,25,26]. The relationships between muscular imbalances, limited mobility, and clinical pain symptoms seem more than logical in the stomatognathic system; however, these obvious relationships are difficult to prove clinically, as the source of all symptoms may not be identified easily. The majority of clinical studies on the validity, sensitivity, and utility of sEMG in TMD diagnosis conclude advantages for the diagnosis with moderate to good sensitivity [27,28,29], but at the same time urge the use of indices to avoid overlap of temporalis and masseter muscles during conduction [30]. On the other hand, Manfredini et al. found no significant differences between sEMG derivations of TMD patients and controls in their studies and questioned at the same time the validity of kinesiographs, which should be used as an alternative [31]. However, with regard to kinesiographs, Cooper [32] provided conclusive refutations of these findings. In contrast to sEMG derivations, ultrasound axiographies are found to be almost invariably positive, valid, sensitive, and beneficial for use in TMD therapy in the relatively small body of literature [33,34,35,36,37,38]. In this context, the results of Stiesch-Scholz et al. [35], who investigated the reproducibility of functional movements of TMD patients using ultrasound facebow technology, appear particularly interesting. In addition to a high accuracy of the measuring system (ARCUS Digma II, KaVo Dental GmbH, Biberach, Germany), they found significantly lower reproducibility of functional movements in TMD patients. Taking these findings into account, the qualitative impressions of the post-treatment function reports can also be reconciled in the present study, whose degree of harmonization visually reflects precisely this increase in reproducibility. With regard to the increased range of motion, Wang et al. investigated the difference in prognathic patients before and after osteotomy, and came to the conclusion that mandibular mobility was significantly lower in dysgnathic patients than in the controls, and that the patients no longer differed significantly from the controls after orthognathic surgery, i.e., an increase in mobility was also observed here [34]. The therapeutic approaches of orthognathic therapy and splint therapy are fundamentally different, but they seem to share the therapeutic effect of increased mobility, which is regarded as a treatment success, and also seems to be associated with a decrease in pain. Considering the difficult quantifiability of TMD, which is always a limitation for clinical studies due to its multi-dimensionality and the small patient cohort, the findings obtained should nevertheless be classified as meaningful and valuable, since on the one hand, they can be placed in the global context, and on the other hand, no comparable study can be found in the current literature.

## 5. Conclusions

The results of the present study indicate the positive effects of Michigan splint therapy, which was able to achieve substantial relief of myofascial pain in the majority of cases within a relatively short treatment period. The applied ultrasound facebow technique proved to be a valuable addition for both the study objective and for visualization on the patient and the confirmation of the diagnosis.

## Figures and Tables

**Figure 1 healthcare-10-01070-f001:**
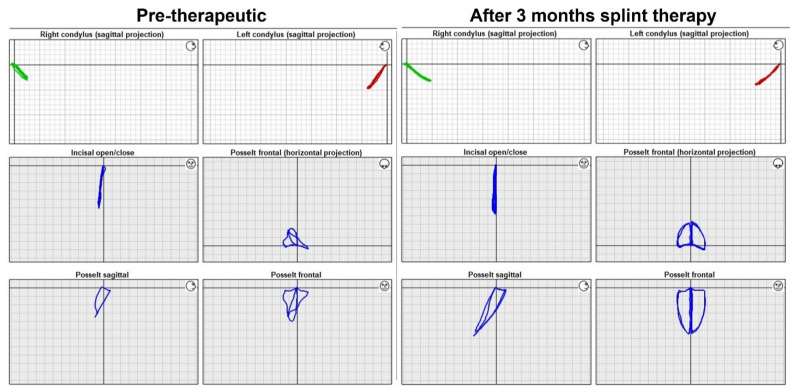
Function report pre- and post-treatment. Indication of right and left condyle in sagittal projection, incisal open/close movements in frontal projection, and Posselt diagram in horizontal, sagittal, and frontal projection before (**left**) and after (**right**) treatment.

**Table 1 healthcare-10-01070-t001:** Median differences of variables tested.

Difference of Variables in (mm)	Median	Min.	Max.	IQR	*p*-Value	Sign
Maximum opening	+6.15	0.20	17.01	7.95	<0.001	***
Left condyle motion during maximum opening	+1.93	−1.29	10.55	3.29	0.191	n.s.
Right condyle motion during maximum opening	+2.73	−0.75	18.00	4.64	0.045	*
Left condyle motion during retrusion	+0.38	−1.15	2.29	0.38	0.716	n.s.
Right condyle motion during retrusion	+0.28	−0.06	1.19	0.55	0.356	n.s.
Left laterotrusion	+0.73	−1.05	4.73	2.54	0.154	n.s.
Right laterotrusion	+0.63	−3.90	5.11	2.12	0.510	n.s.

Wilcoxon test: n.s. = not significant, * = *p* < 0.05, *** = *p* < 0.001, significance level = *p* < 0.05.

**Table 2 healthcare-10-01070-t002:** Qualitative assessment of movement harmonization.

Patient	Right Condyle (Sagittal)	Left Condyle (Sagittal)	Maximum opening (Frontal)	Posselt (Frontal)	Posselt (Sagittal)	Posselt (Axial)
**001**	**(+)**	**(+)**	**+**	**+**	**+**	**+**
**002**	**+**	**+**	**+**	**+**	**+**	**+**
**003**	**(+)**	**(+)**	**+**	**+**	**(+)**	**+**
**004**	**+**	**+**	**+**	**+**	**+**	**+**
**005**	**+**	**+**	**+**	**+**	**+**	**+**
**006**	**+**	**(+)**	**(+)**	**+**	**+**	**+**
**007**	**+**	**+**	**+**	**(+)**	**(+)**	**(+)**
**008**	**(+)**	**+**	**+**	**+**	**+**	**+**
**009**	**+**	**(+)**	**+**	**(+)**	**+**	**(+)**
**010**	**–**	**–**	**(+)**	**(+)**	**(+)**	**(+)**
**011**	**–**	**–**	**+**	**+**	**+**	**+**
**012**	**(+)**	**+**	**(+)**	**+**	**+**	**+**
**013**	**(+)**	**–**	**–**	**–**	**(+)**	**–**
**014**	**+**	**+**	**+**	**(+)**	**(+)**	**(+)**
**015**	**(+)**	**(+)**	**+**	**–**	**+**	**–**
**016**	**+**	**+**	**+**	**+**	**+**	**+**
**017**	**–**	**–**	**+**	**(+)**	**+**	**(+)**
**018**	**+**	**+**	**+**	**+**	**+**	**+**
**019**	**+**	**+**	**+**	**+**	**+**	**(+)**
**020**	**–**	**–**	**(+)**	**–**	**+**	**(+)**
**021**	**+**	**+**	**+**	**+**	**+**	**+**
**022**	**–**	**–**	**+**	**+**	**(+)**	**(+)**
**023**	**(+)**	**(+)**	**+**	**+**	**–**	**–**
**024**	**(+)**	**–**	**+**	**–**	**–**	**–**
**025**	**+**	**+**	**+**	**+**	**+**	**+**
**026**	**+**	**(+)**	**+**	**(+)**	**–**	**–**
**027**	**–**	**–**	**(+)**	**(+)**	**(+)**	**(+)**
**028**	**+**	**+**	**(+)**	**+**	**(+)**	**+**
**029**	**(+)**	**–**	**(+)**	**–**	**(+)**	**–**

“+” indicates a qualitative and a quantitative improvement, “(+)” indicates a solitary improvement in the degree of harmonization (qualitative) or in the range of motion (quantitative), “–” indicates that neither a qualitative nor quantitative improvement was observed.

**Table 3 healthcare-10-01070-t003:** Qualitative assessment of pain perception.

Number of Patients	No Change from Baseline	Improvement from Baseline	Complete Remission	Exacerbation from Baseline
29	5 (17)	24 (83)	17 (59)	0 (0)

The corresponding patient distribution is presented in n and (%).

## Data Availability

Not applicable.

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
