# Peer review of "Changes in Maximum Mandibular Mobility Due to Splint Therapy in Patients with Temporomandibular Disorders"

_healthcare, 2022, doi:10.3390/healthcare10061070_

Round 1

Reviewer 1 Report

This is a literature about the quantitative and qualitative evaluation of the patients with myofascial pain after 3 months of Michigan splint treatment.

It is necessary to describe the full character of the abbreviation in first time.

L.125; MIP, L.241; sEMG, L.69; tmd (in capital)

The description of the verbal analog scale in four grade in Material and Methods (L.137) disappear in Results and not mentioned in Discussions.

The evaluation method of the harmonization (L.191) is not described in Materials and Methods.

The reference is necessary about “Posselt’s envelope of motion” (L. 195).

It is necessary to explain about the colors, i.e. white, light gray and bold gray in the legend of the table 2.

The reference No. 17 in line 50 is inadequate to explain about the concept of the Michigan splint.

In reference list, the name of the journal of the reference No.27 is “J Electromyogr Kinesiol.

Author Response

Detailed response to reviewer                                                                       healthcare-1756565

To Reviewer #1: Dear Reviewer #1 We would like to sincerely thank you for your efforts and endorsement of our submitted manuscript. We take the liberty to assure you that we have implemented all concerns in the manuscript to the best of our knowledge and belief, as well as made use of professional editing by MDPI. Please allow us to address the editing of your comments afterwards.

To Reviewer #1 Comments

It is necessary to describe the full character of the abbreviation in first time. L.125; MIP, L.241; sEMG, L.69; tmd (in capital)

Author response: Dear Reviewer #1, thank you very much for your valuable advice. We have taken the liberty of revising the text passages accordingly.

“The description of the verbal analog scale in four grade in Material and Methods (L.137) disappear in Results and not mentioned in Discussions.”

Author response: Dear Reviewer #1, thank you very much for your important advice. We have taken the liberty to supplement the results with another subheading and an associated table. Further we addressed the results in the discussion.

“The evaluation method of the harmonization (L.191) is not described in Materials and Methods.”

Author response: Dear Reviewer #1, please forgive us for not going into detail about this aspect. We have allowed to implement a corresponding text passage.

“The reference is necessary about “Posselt’s envelope of motion” (L. 195).”

Author response: Dear Reviewer #1, thank you very much for your valuable advice. We have taken the liberty to implement the corresponding original reference.

“It is necessary to explain about the colors, i.e. white, light gray and bold gray in the legend of the table 2.”

Author response: Dear Reviewer #1, thank you very much for your comment. Originally, we had included the explanation in the body text, but for layout reasons we decided against a color subdivision and removed the grayscale.

“The reference No. 17 in line 50 is inadequate to explain about the concept of the Michigan splint.”

Author response: Dear Reviewer #1, thank you very much for your valuable advice. We have taken the liberty of adding the original publication.

Reviewer 2 Report

Dear Authors,

The aim of the present study was therefore to evaluate the therapeutic effect of Michigan splints qualitatively in terms of pain reduction and quantitatively in terms of motion-range-improvement in patients diagnosed with myofascial pain (diagnosis I.a and I.b) according to the RDC/TMD criteria

Authors concluded reporting the positive effects of Michigan splint therapy, which was able to achieve substantial relief of myofascial pain in the majority of cases within a rather short treatment period.

Studies that select only muscle or joint only patients are few in the literature. Although the study is not a randomized controlled trial, the results are of clinical interest and in line with the aims of the journal. The author guidelines have been respected. 

The Result and Discussion sections were well described. 

References were well reported.

However, there are some issues that should be addressed. 

Abstract

(The abstract should be a total of about 200 words maximum. The abstract should be a single paragraph and should follow the style of structured abstracts, but without headings: 1) Background: Place the question addressed in a broad context and highlight the purpose of the study; 2) Methods: Describe briefly the main methods or treatments applied. Include any relevant preregistration numbers, and species and strains of any animals used. 3) Results: Summarize the article's main findings; and 4) Conclusion: Indicate the main conclusions or interpretations. The abstract should be an objective representation of the article: it must not contain results which are not presented and substantiated in the main text and should not exaggerate the main conclusions.)

See Instruction for author https://www.mdpi.com/journal/healthcare/instructions

Introduction

Line 39. Which difference between arthrosis and osteoarthrosis? Do you mean arthrosis and arthritis?

Line 40. “research diagnostic criteria for temporomandibular disorders”. Please write as “

 Research Diagnostic Criteria for Temporomandibular Disorders (RDC/TMD).

Lines 44-45. A recent systematic review with meta-analysis reported the efficacy of conservative approaches in TMD of only muscular origin. Please report treatments showed to be efficacy, and cite this review - Efficacy of rehabilitation on reducing pain in muscle-related temporomandibular disorders: A systematic review and meta-analysis of randomized controlled trials.

Materials and Methods

When patients were recruited? From..to?

Why do you use RDC/TM and not DC/TMD (Shiffman 2014)?

Only therapy with occlusal splint was made? No behavioral therapy)

How many hours/day? 

Sometimes p-value was written as “p” and sometimes as “P”. Please use the same form.

The Result and Discussion sections were well described. 

References were well reported. 

Author Response

Detailed response to reviewer                                                                       healthcare-1756565

To Reviewer #2: Dear Reviewer #2 We would like to sincerely thank you for your efforts and endorsement of our submitted manuscript. We take the liberty to assure you that we have implemented all concerns in the manuscript to the best of our knowledge and belief, as well as made use of professional editing by MDPI. Please allow us to address the editing of your comments afterwards.

To Reviewer #2 Comments

(The abstract should be a total of about 200 words maximum. The abstract should be a single paragraph and should follow the style of structured abstracts, but without headings: 1) Background: Place the question addressed in a broad context and highlight the purpose of the study; 2) Methods: Describe briefly the main methods or treatments applied. Include any relevant preregistration numbers, and species and strains of any animals used. 3) Results: Summarize the article's main findings; and 4) Conclusion: Indicate the main conclusions or interpretations. The abstract should be an objective representation of the article: it must not contain results which are not presented and substantiated in the main text and should not exaggerate the main conclusions.)

See Instruction for author https://www.mdpi.com/journal/healthcare/instructions

Author response: Dear Reviewer #2, we thank you very much for your appreciation and critical consideration of the manuscript and hope that the changes made to the abstract have met your expectations.

“Line 39. Which difference between arthrosis and osteoarthrosis? Do you mean arthrosis and arthritis?”

Author response: Dear Reviewer #2, thank you very much for your thoughtful comment. We have followed your recommendation and revised the text accordingly.

Patients were treated by the Department of Prosthodontics at the University Medical Center Göttingen and recruited between July 2014 and March 2016. “Lines 44-45. A recent systematic review with meta-analysis reported the efficacy of conservative approaches in TMD of only muscular origin. Please report treatments showed to be efficacy, and cite this review - Efficacy of rehabilitation on reducing pain in muscle-related temporomandibular disorders: A systematic review and meta-analysis of randomized controlled trials.”

Author response: Dear Reviewer #2, thank you very much for your reference to this wonderful work. We have taken the liberty of following your recommendation and placed the review in the core of our introduction.

“When patients were recruited? From..to?”

Author response: Dear Reviewer #2, thank you very much for your comment. The study was conducted as a part of a larger research project (founded by the German Research Foundation) in which tmd patients and tmp anatomy/movements have been evaluated by real-time-MRI, the project began 2014 and ended 2018, the clinical experiments from this manuscript are, of course, previously unpublished. We took the liberty of implementing the following text passage:

Revised textpassage: Patients were treated by the Department of Prosthodontics at the University Medical Center Göttingen and recruited between July 2014 and March 2016.

“Why do you use RDC/TM and not DC/TMD (Shiffman 2014)?”

Author response: Dear Reviewer #2, thank you very much for your valuable question. DC/TMD have been published 2014, when we had just started the general research project (please see above), therefore theses specific parameters could not be changed during the running project. RDC-TMD were only used to classify the patients into the diagnostic group of „myofacial pain“, therefore we consider this aspect to be more neglected, also because the RDC-TMD are used as reference in many studies until today.

“Only therapy with occlusal splint was made? No behavioral therapy?”

Author response: Dear Reviewer #2, thank you very much for your valuable question. Exactly. Only splint therapy was studied to obtain direct conclusions about the therapeutic agent and no additional effects of adjuvant therapies.

“How many hours/day?”

Author response: Dear Reviewer #2, thank you very much for your valuable question. We have asked patients to use the splints exclusively for nighttime application and have marked this accordingly in the manuscript. Since an average sleep duration of 7-8 hours could be assumed, the application should have been concentrated over this period per day. However, we could not verify this.

“Sometimes p-value was written as “p” and sometimes as “P”. Please use the same form.”

Author response: Dear Reviewer #2, we apologize for this messy writing and at the same time assure you that we have thoroughly revised this aspect.

“The Result and Discussion sections were well described. References were well reported.”

Author response: Dear Reviewer #2, finally, please allow us to thank you once again for your detailed and critical examination of our submitted work.